Long-term changes in soil biological activity and other properties of raised beds in Longan orchards

Nguyen Nghia Khoi nknghia@ctu.edu.vn 1
Nguyen Phuong Minh 1
Chau Anh Thy Thi 1
Do Luan Thanh 1
Nguyen Thu Ha Thi 1
Tran Duong Hai Vo 2
Le Xa Thi 3
Robatjazi Javad 4
Lasar Hendra Gonsalve W. 4
Morton Lois Wright 5
Demyan M. Scott 6
Tran Huu-Tuan 7
Tecimen Hüseyin Barış 8
1 Department of Soil Science, College of Agriculture, Can Tho University , Can Tho City , Can Tho , Vietnam
2 Department of Agriculture and Aquaculture, Bac Lieu Technical and Economic College , Bac Lieu City , Bac Lieu , Vietnam
3 School of Education, Soc Trang Community College , Soc Trang City , Soc Trang , Vietnam
4 Department of Soil and Crop Sciences, Texas A&M University , College Station , TX , United States of America
5 Department of Sociology and Criminal Justice, College of Agriculture and Life Sciences, Iowa State University , Ames , IA , United States of America
6 School of Environment and Natural Resources, Ohio State University , Columbus , OH , United States of America
7 Science and Technology Advanced Institute, Van Lang University , Ho Chi Minh City , Ho Chi Minh , Vietnam
8 Department of Environmental Sciences, College of Agriculture, Tennessee State University , Nashville , United States of America
Singh Anshuman
Electronic publication date: 2024 Nov 6
Publication date: 2024
Volume: 12
Electronic Location ID: e18396
Received 2024 Jul 8; Accepted 2024 Oct 3
Copyright: ©2024 Nguyen et al.
Copyright year: 2024
Copyright holder: Nguyen et al.
License: This is an open access article distributed under the terms of the Creative Commons Attribution License, which permits unrestricted use, distribution, reproduction and adaptation in any medium and for any purpose provided that it is properly attributed. For attribution, the original author(s), title, publication source (PeerJ) and either DOI or URL of the article must be cited.
License URL: https://creativecommons.org/licenses/by/4.0/

Keywords: Enzyme activity, Fertility, Fruit trees, Microorganisms, Sustainable management

Funding: Ministry of Education and Training, Vietnam B2023–TCT-14 This work was supported by the Ministry of Education and Training, Vietnam, Grant Number: B2023–TCT-14. The funders had no role in study design, data collection and analysis, decision to publish, or preparation of the manuscript.

==============================
Introduction

The Longan fruit tree of the Vietnam Mekong Delta is grown in raised beds to improve water drainage during the rainy season and can live as long as 100 years.

Objective

This research explores the extent to which the soil microorganisms as well as soil physical and chemical properties of these raised beds degrade over a period of 60 years under traditional management practices.

Materials and Methods

Raised bed topsoil samples at depths of 0–20 cm were obtained from four different Longan orchards raised bed age groups: group 1) 15–25 years (L1–L5); group 2) 26–37 years (L6–L10); group 3) 38–45 years (L11–L15); and group 4) 46–60 years. Soil biological properties were tested for nitrogen-fixing bacteria, phosphorus solubilizing bacteria, potassium solubilizing bacteria, calcium solubilizing bacteria and silicate solubilizing bacteria, β-glucosidase, urease, phosphomonoesterase, and phytase. Soil samples were also tested for moisture content, soil texture, soil porosity, and bulk density as well as soil chemical properties including pH, electrical conductivity (EC), soil organic matter (SOM), total nitrogen (TN), total phosphorus (TP), total potassium (TK), available nitrogen (NH4+, NO3−), available phosphorus (AP), exchangeable potassium (K+), exchangeable calcium (Ca2 +), available silicate (SiO2), available copper (Cu), zinc (Zn), boron (B) and manganese (Mn). Key findings: The results showed that soil moisture, soil porosity, sand content, SOM, TP, TK, available P, exchangeable Ca2 +, available Si, nitrogen fixing bacteria number, β-glucosidase, urease, phosphomonoesterase, and phytase gradually and significantly decreased in the raised bed soil as the Longan orchard increased in age. Pearson correlation analysis between the ages of Longan orchards and soil properties revealed that raised bed ages were positively correlated with soil bulk density, but negatively correlated with soil moisture content, soil porosity, SOM, TN, β-glucosidase, urease, phosphomonoesterase, and phytase. Principal component analysis (PCA) showed Longan yields had a positive correlation with available NO3− but negative correlation with NFB, exchangeable Ca2 +, pH, and available B. These findings reveal that traditional long-term management of Longan trees in raised beds significantly reduce soil organic matter, moisture content, porosity, and soil fertility with impacts on soil microbial numbers and activity within raised bed soils.

Future Directions

This suggests that more sustainable management practices, such as mulch and cover crops that decrease soil compaction and increase soil organic matter, improve soil porosity, total N, and feed soil microorganisms that are critical to nutrient cycling are needed to improve raised bed soil quality.

Introduction

The Vietnam Mekong Delta (VMD) has an area of about 4.0 million ha, including over 2.4 million ha of agricultural land and plays a key role in producing over 70% of fruit for the whole country (General Statistics, Office of Vietnam, 2020). The Longan (Dimocarpus Longan Lour) tropical tree fruit, used both fresh and dried for food and medicinal purposes has high nutritional and economic value with strong market demand beyond Asia as an exotic fruit (General Statistics Office of Vietnam, 2021; Thanh Truc & Thuc, 2022). The Longan farmers in the VMD use raised beds to grow Longan in their orchards to avoid seasonal flooding and waterlogged conditions in the low-lying coastal regions (Quang, 2013; Thao, Takagi & Esteban, 2014; Dang & Hung, 2023; Morton, Nguyen & Demyan, 2023). These raised soil beds, built from sediments excavated from adjacent lateral ditches, rise 0.6 to 0.7 m above the ditches to improve drainage and leach toxic minerals in acid sulfate soils away from the tree root zone (Morton, Nguyen & Demyan, 2023; Dang & Hung, 2023). The monsoon tropical precipitation regime in the Delta has high humidity and heavy rains in the wet season, lasting from May to November (Tran, Menenti & Jia, 2022). High average annual precipitation (1,370–2,394 mm) causes waterlogging, poor drainage, and crop production decline (Manik et al., 2019; Adegoye et al., 2023). Fruit trees exposed over time to waterlogging conditions have increased root rot caused by soil pathogens that reduce tree growth and productivity. Many commercial orchards rely on raised beds and intensive management regimes such as synthetic fertilizers, fertigation, and chemical pesticides to maintain drainage and soil fertility (Webber et al., 2022).

The fertility of fruit orchard soils has significantly declined in recent decades (Zhao et al., 2020; Dang, Ngoc & Hung, 2021; Dang, Ngoc & Hung, 2022). Key factors contributing to soil fertility reduction and degradation are compaction, soil organic matter loss, salinization, nutrient depletion, and pollution as a result from a long-term use of inorganic fertilizers (Duan et al., 2016; Van & Ngoc, 2020). Intensive farming practices involving heavy chemical applications contribute to soil degradation and reduced soil fertility in orchard soils of the VMD. Under tropical conditions, exchangeable cations are easily removed from the soil surface by rainwater and irrigation, resulting in increased soil acidification and compaction (Natale et al., 2012; Dang & Hung, 2023). Continual overuse of chemical fertilizers and pesticides has led to decline in soil organic matter and fertility and increased soil acidification (Wang et al., 2020). The application of fertilizer directly on soil influences soil microbial diversity and soil nutrient levels (Bell et al., 2015). Repeated overuse of agrochemicals has a detrimental impact on soil quality, soil health, and soil microbial community structure primarily due to a significant reduction in soil pH, leading to decreased bacterial diversity and alterations in bacterial community composition.

Long-term tropical fruit orchard cultivation with traditional soil management practices in some regions has been found to lead to changes in the soil’s physical, chemical and biological properties, in addition to reduced productivity due to insects, diseases, and unsuitable soil characteristics (Xiang et al., 2018). These studies report that soil exposed to various degradation processes have changes in soil fertility, organic carbon, alkalization, and salinization with impacts on soil quality, fruit yields and quality (Xu et al., 2019; Chen & Xu, 2008; Maleki et al., 2021). Lands for cultivating perennial fruit trees in the Mekong River Delta region are currently facing declining pH levels, low nutrient content and organic matter, imbalanced nutrition, and reduced microbial activity as the orchards age (Quang & Guong, 2011; Thiet & Guong, 2014). Vinh Long, which has the largest area of Longan cultivation in the VMD (6,129 ha) accounting for 24.6% of the total area of Longan cultivation, is experiencing many of these issues. Due to the large-scale and long-term application of chemical fertilizers and pesticides, the Vinh Long region is facing secondary salinization, decreased soil fertility, and increasing environmental pollution (Hung et al., 2019).

There is a need to better understand the extent to which soil quality- physical, chemical and biological properties of orchard raised beds degrade over many years under traditional management practices. This study hypothesizes that long-term raised bed soil management practices in VMD Longan orchards have significantly led to changes in soil microbial and geo-chemical properties over a period of 60 years. Of particular interest are changes in biological parameters and decline in soil exchangeable cations, porosity, and fertility of the surface and subsurface soils as trees grown in raised bed mature and age. Although a couple studies have looked at orchard cultivation, erosion, and soil physical and chemical properties (Xiang et al., 2018; Dang & Hung, 2023), none to our knowledge have analyzed soil biological properties including soil enzyme activity of Longan raised beds over long periods of time and integrated these findings with soil physical and chemical properties of raised beds in tropical orchards. First materials, methods and descriptions of the study sites are presented, followed by the results of soil physical and chemical properties of 20 VMD orchards and their biological parameters. A correlation matrix and principal component analysis (PCA) are used to analyze relationships among the soil biological and geo-chemical properties of the Delta Longan orchards’ raised bed soils. Then findings from this study are used to suggest future research and practical management recommendations to improve soil biological functioning and other soil properties that can improve the Longan fruit tree production system.

Materials and Methods

Site description

Longan growing areas in Vietnam are concentrated in a number of provinces in the North and Mekong River Delta provinces. Statistical data in 2017 show the whole country had about 73.3 thousand ha planted to trees with production estimated at 550,000 t year−1 (Hung et al., 2019). In the Mekong River Delta provinces, the total Longan growing area is 24,913 ha, achieving a productivity of 227,624 tons, and accounting for 40% of the total yield of the country. Of these, four regions have the largest Longan growing areas in descending order as follows Vinh Long (6,129 ha), Dong Thap (5,515 ha), Soc Trang (3,552 ha), and Can Tho (2,512 ha) (General Statistics, Office of Vietnam, 2020). The most common productive cultivars grown in the Mekong River Delta provinces include “Tieu da bo”, “Xuong com vang”, and “Xuong com rao” with a mean productivity of 10 t year−1. Traditional production systems use intensive and conventional chemical farming techniques to increase fruit production to 20–25 t year−1. Further, off-season flowering treatment techniques to control harvesting time are regularly used in the Longan growing areas in the Mekong River Delta (Hung et al., 2019).

The study area is located in An Binh commune, Long Ho district, Vinh Long province (10°17′37.5″N 105°58′55.7″E). This region has a remarkable historical background for Longan cultivation in Vinh Long and in the VMD as well. The study site has an altitude 0.6−1.2 m above sea level; receiving 1,450–1,504 mm year−1 annual precipitation with a flooding season between September and November (Thanh et al., 2019). The Mekong River’s influence during the Holocene retrograde marine sedimentation processes left behind marine sandy soils and saline soils from saltwater intrusion (Ta et al., 2021). The dominant climate in the region has been defined as tropical climate with monsoon. Twenty soil samples from different ages of Longan orchard beds cultivated longer than 15 years were collected and analyzed. Information regarding the age of the orchards and bed ages are shown in Table S1.

Soil sample collection

Soil samples of Longan orchards in Vinh long province, Vietnam were collected in July, 2023 after the final Longan harvest with a total of 20 soil samples from four different groups of Longan orchards based on age of the raised bed. Samples were grouped by the age of the raised bed: Group 1) 15–25-year-old Longan orchards (L1–L5); group 2) 26-37-year-old Longan orchards (L6–L10); group 3) 38–45-year-old Longan orchard (L11–L15); and group 4) 46–60-year-old Longan orchards (L16–L20). Each raised bed age group consisted of soil drawn from five different Longan orchard farms corresponding to five different soil samples. Due to budget limitations, only 20 soil samples were collected for this study. Twenty is considered a sufficiently large enough sample to be representative of the area of study area. At each Longan orchard farm, soil samples were collected using soil auger (Seymour 21306 AU-S6, USA) from the upper layer depth of 0–20 cm, under the canopy of Longan trees from 10 different spots within each Longan farm. The soil was combined to constitute one soil sample from several locations in a zig-zag pattern for homogeneity and kept in clean nylon bags. Then the samples were transported in hard plastic boxes containing liquid nitrogen from the field to the laboratory to preserve the original physical structure of the soil. In the lab, the collected fresh soil samples were divided into two subsamples. One subsample was used to analyze immediately soil microbiology and enzyme activities including the numeration of beneficial bacteria such as nitrogen-fixing bacteria, phosphorus solubilizing bacteria, potassium solubilizing bacteria, calcium solubilizing bacteria and silicate solubilizing bacteria and soil enzyme activities such as β-glucosidase, urease, phosphomonoesterase, and phytase. The other subsample was air-dried at 4 °C so that the soil moisture of all samples reached 10–12%. Then, the air-dried sample was ground through a 0.5-2 mm sieve (G.I. Sieves, Rachana, India) to homogenize the sample as well as to remove the roots and plant debris and used for analysis of soil physical and chemical properties including moisture content, soil texture, soil porosity, and bulk density. Soil chemical properties included tests for pHw, electrical conductivity (EC), soil organic matter (SOM), total nitrogen (TN), total phosphorus (TP), total potassium (TK), available nitrogen (NH4+, NO3−), available phosphorus (AP), exchangeable potassium (K+), exchangeable calcium (Ca2+), available silicate (SiO2) and some trace elements such as copper (Cu), zinc (Zn), boron (B) and manganese (Mn).

Soil physical property determination

Soil bulk density was determined using undisturbed soil samples that were collection using a soil auger equipped with Royal Eijkelkamp soil sample rings with a volume of 98.125 cm3 (C53; Royal Eijkelkamp, Giesbeek, The Netherlands). Soil was dried in a 105 °C oven for 24 h and bulk density calculated as the mass of oven-dried soil divided by the total volume (Mtyobile, Muzangwa & Mnkeni, 2020; Van, Ngoc & Hung, 2022). Soil porosity was then computed by dividing the soil bulk density to soil particle density (Mtyobile, Muzangwa & Mnkeni, 2020). The soil porosity calculation formula is shown in the following equation: Soil porosity=1−Soil bulk density/Soil particle density∗100

Soil texture was determined according to the sieve (sand particles (0.02–2 mm); silt (0.002–0.02 mm) and clay (<0.002 mm) method. Robinson’s pipette method was carried out on the basis of Stoke’s law (Grossman & Reinsch, 2002).

Soil chemical property determination

Soil pHw and EC were measured in a prepared soil slurry suspension at a ratio of 1:2.5 (w/v) (soil:water) with a pH and EC meter (Thomas, Haszler & Blevins, 1996). Soil organic matter was determined by Walkley-Black method (Nelson & Sommers, 1982). Total nitrogen was determined using the Kjeldahl distillation method which converts all organic nitrogen in the sample to N-NH4+ form (Horneck & Miller, 1998). Soil samples were digested with H2SO4 and HClO4 to determine total phosphorus and total potassium. Then, the sample was mixed with phosphomolybdate using the reducing agent ascorbic acid and measured colorimetrically on a spectrophotometer (Shimadzu-UV–VIS UVmini-1240, Japan) at 880 nm wavelength. Total soil potassium was measured using an atomic absorption spectrometer and a flame photometer (Shimadzu AA-7000 Series Consumables, Shimadzu, Tokyo, Japan) (Jackson, 1958). Soil N-NO3− and N-NH4+ were extracted by 2M KCl. The N-NH4+ content is determined by showing the formation of a blue compound due to the reaction between ammonium and salicylate and hypochlorite ions in the presence of sodium nitroprusside. The N-NO3− concentration was measured by converting NO3− to NO2− in a VCl3 solution. In an acidic environment, the NO2− solution combines with sulfanilic acid and α-naphthylamine reagents to produce a pink solution that can be detected by spectrophotometer, at 540 and 650 nm wavelengths, respectively (Shimadzu-UV–VIS UVmini-1240; Shimadzu, Tokyo, Japan) (Bremner, 1996). Soil available phosphorus was determined according to the Olsen method (Olsen & Sommers, 1982). Soil exchangeable potassium and Ca2+was extracted with 0.1M BaCl2 and the elemental content was measured on an atomic absorption spectrometer (Shimadzu AA-7000 Series Consumables) (Thomas, 1982). Soil available boron was dissolved in hot distilled water to react with Azomethine H in a buffer solution to create a yellow mixture, then the sample was measured on a spectrophotometer at 420 nm wavelength (Shimadzu AA-7000 Series Consumables) (Wolf, 1971). Soil soluble silicon content was determined on the UV-Visible spectrophotometer (Shimadzu-UV–VIS UVmini-1240) by the silicon molybdenum blue reaction between a wavelength of 400 nm and 800 nm (Hallmark, Wilding & Smeck, 1982). Trace elements such as total soil Copper, Zinc, and Manganese were determined according to the Mehlich 3 method (Mehlich, 1984).

Soil biological property determination

Numeration of soil beneficial bacterial density

The density of beneficial bacteria including nitrogen-fixing bacteria, phosphorus-solubilizing bacteria, silicon-solubilizing bacteria, calcium-solubilizing bacteria, and potassium-solubilizing bacteria was determined according to the method of Gerba & Pepper (2004). Phosphate saline buffer (PSB) solution was used to extract the samples for one hour at 150 rpm, with a ratio of 1:10 (w/v) soil: PSB. The suspension extraction was spread on Burk’s nitrogen-deficient medium, NBRIP, soil extract agar (SEA), Devenze-Bruni (DB), and Aleksandrov to determine the density of nitrogen fixing bacteria, phosphorus solubilizing bacteria, silicon solubilizing bacteria, calcium solubilizing bacteria and potassium-solubilizing bacteria, respectively, then the sample plates were incubated at 30 °C for 3–5 days (Mehta & Nautiyal, 2001; Hu, Chen & Guo, 2006; Vasanthi et al., 2013; Cacchio et al., 2014).

Soil enzyme activities

Soil phosphomonoesterase enzyme activity was determined according to the method of Tabatabai & Bremner (1969) and Eivazi & Tabatabai (1977). Briefly, a subsample of 1 g soil (dry weight) was heated at 37 °C for 1 h in four mL modified universal buffer, 0.25 mL toluene and one mL p-nitrophenol phosphate (15 mM). After 1-h incubation, one mL of 0.5 M CaCl2 and four mL of 0.5 M NaOH were added to the mixture. The soil suspension was mixed well and filtered through a Whatman No. 1005-110 filter paper. The optical density of the filtrate was measured under the absorbance at 400 nm by a spectrophotometer (Shimadzu-UV–VIS UVmini-1240, Japan).

Soil β-Glucosidase enzyme activity was also determined through the amount of p-nitrophenol released after soil incubation with p-nitrophenyl glucoside. Briefly, one g moist, sieved (two mm) soil (dry weight) was placed in an Erlenmeyer flask, then 0.25 mL toluene, four mL modified universal buffer solution, and one mL ρ-Nitrophenyl- β-D-glucoside 0.05M solution were added to the flasks. The flasks were stoppered and contents mixed thoroughly and then incubated for 1 h at 37 °C. After the incubation, one mL CaCl2 0.5M solution and four mL of Tris buffer 0.1M were added to the flasks, mixed well and the soil suspensions were filtered immediately with Whatman No. 1005-110 filter paper. The soil suspension was measured under the color intensity at 400 nm by a spectrophotometer (Shimadzu-UV–VIS UVmini-1240) (Tabatabai, 1982; Eivazi & Tabatabai, 1988).

Soil urease enzyme activity was determined according to the improved method of Cordero, Snell & Bardgett (2019) and Kandeler & Gerber (1988) through colorimetric determination of released ammonia (at 650 nm) on the UV-Visible spectrophotometer (Shimadzu-UV–VIS UVmini-1240) after the incubation of soil sample with urea solution for 2 h at 37 °C. Briefly, 2.5 g soil (dry weight) was placed in Erlenmeyer flasks and moistened with 1.25 mL 0.08 M Maqueous urea solution, then stoppered and placed in an incubator at 37 °C. After 2 h incubation the stoppers were removed, then a 50 mL mixture containing 1:1 (v/v) ratio of 1N KC1 and 0.01N HCl was added to the flasks. The samples were shaken on a mechanical shaker (GFL 3018 Shaker-3018, Germany) for 30 min, then centrifuged at 2,900 g for 5 min. Then 75 µL supernatant was pipetted into 96 transparent well plates and mixed with 75 µL water. Ammonia concentration was evaluated by Berthelot reaction (Krom, 1980) using two different reagents: (1) an oxidation solution (1 mg mL−1 dichloroisocyanuric acid sodium salt dehydrate) and (2) a color reagent comprised of 2:1 (v:v) ratio solution of 0.15 M NaOH and a solution containing 170 mg mL −1 sodium salicylate and 1.278 mg mL−1 sodium nitroprusside dehydrate (prepared just before usage). For ammonia measurement, each well received 75 µL of color reagent followed by 30 µL of the oxidation solution. Each well was properly mixed by pipetting and the color was evaluated after 30 min under the absorbance at 650 on the UV-Visible spectrophotometer (Shimadzu-UV–VIS UVmini-1240).

Soil phytase enzyme activity was determined according to the improved method of Shimizu (1992) and Berry, Shang & Zelazny (2009). Briefly, a subsample of 1 g soil (dry weight) was heated at 35 °C for 1 h in four mL acetate buffer, 0.2 mL toluene and two mL sodium phytate (7.61 mM). Samples were then centrifuged at 2,900 g for 5 min and the liberated inorganic orthophosphate (Pi) was determined under the absorbance at 700 nm on the UV-Visible spectrophotometer (Shimadzu-UV–VIS UVmini-1240).

Statistical analysis

One-way analysis of variance (ANOVA) was used in data analysis. Results are presented as means ± standard deviation (SD), and significant differences were verified by computing the Tukey’s Test at the level of 5%. Statistical tests were conducted with the Minitab (version 16.2) software. In R (R Core Team, 2023), a principal component analysis (PCA) was performed using the built-in facto extra package from the replicated data. The suitability of data for PCA was tested by using the Bartlett test after standardization. Pearson’s correlation was conducted using the ‘cor’ function and a correlation matrix was generated using the ‘corrplot’ package.

Results and Discussion

Soil physical properties of different Longan orchards

The majority of soil samples collected from various Longan orchards representing different raised bed ages exhibited soil textures in the silty clay and silty clay loam classes, as depicted in the USDA soil textural triangle (Fig. 1 and Table 1). These Longan orchard soils typically contained mean percentages of sand, silt, and clay at 5.7%, 60%, and 35%, respectively. The soil textures ranged from medium to fine-textured soils which are typical for the alluvial soils found found in the middle of the Vietnam Mekong Delta, and are well suited for fruit orchard cultivation (Le, 2003). Additionally, the particle density of these Longan orchard soils ranged from 2.43 to 2.46 g cm−3, within the normal range for mineral soils in the cultivated layer. The soil moisture content in the surface layer of Longan orchards was quite low, ranging from 24.7% to 29.2% as the soil was sampled at harvest time. The soil bulk density of these 20 different Longan orchard soils varied between 1.15 to 1.31 g cm−3, a density which does not affect root growth. The soil porosity of the soil samples was between 46.8% and 53.8%, ranking from low to moderate which is a typical porosity range for clay soil.

Figure 1 Variation of soil texture classes of Longan orchard soils in Vinh Long, Vietnam.

Table 1 Soil physical properties of different Longan orchard soils in Vinh Long, Vietnam.

Raised bed age (Year)	Soil physical properties	
	Moisture content (%)	Bulk density (g cm−3)	Particle density (g cm−3)	Soil porosity (%)	Soil texture (%)	
					Sand	Silt	Clay	
15–25	29.2a	1.15b	2.43	53.8a	8.01a	60.5	31.5b	
26–37	28.4a	1.17ab	2.44	52.1a	4.46b	59.3	36.1a	
38–45	27.3ab	1.21ab	2.44	49.4ab	5.11ab	60.0	34.3a	
46–60	24.7b	1.31a	2.46	46.8b	3.82b	59.1	37.0a	
F	*	*	ns	*	*	ns	*	
CV (%)	9.60	8.43	1.14	7.33	42.7	5.23	7.39	
Notes.

* Means in the same rows with different letters are significantly (p < 0.05) difference according to Tukey’s test.

ns, non-significant; n = 5.

Significant variations were observed for soil bulk density, soil porosity, soil moisture content, sand and clay among orchards grown in raised beds of different ages (Table 1) (p < 0.05). Lower soil moisture content, soil porosity and sand content and significantly higher soil bulk density and clay content were decoupled as the Longan orchard beds increased in age. These variations suggest differing levels of soil compaction, probably the long-term impacts of farming practices such as pest and crop control, irrigation, and soil surface management as observed during soil surveys and local farmer interviews. The significant reduction of soil moisture content, porosity, and sand content with increasing soil bulk density, and clay content over increasing duration of Longan cultivation indicates increased soil degradation and soil compaction. The Longan farmers in this study have applied traditional chemicals using intensive management practices in their orchards including burning all the Longan residues leaving behind a bare soil surface. This practice as well as regular herbicide applications are increasing soil degradation, erosion and compaction (Duan et al., 2020). These conditions are reflected in the loss of soil organic matter. In this study the soil organic matter was significantly reduced with an increase in the orchard’s raised bed age. Many previous studies have found that soil porosity increases as the soil organic matter content increases (Nabayi et al., 2021; Fukumasu et al., 2022; Dang & Hung, 2023). The chemical, physical and biological processes in soil are strongly regulated by soil organic matter (Hoffland et al., 2020; Zhai et al., 2022). Improvements soil organic matter can help to increase soil porosity, water-holding capacity, and aggregate stability (Nabayi et al., 2021). A decrease of soil water content, soil porosity, sand content while increasing soil bulk density, and clay content was observed in the older raised beds and may be attributed to use of unsustainable cultivation practices and no incorporation of additional organic materials into the soil (Dang & Hung, 2023). Moreover, in this current study, we found that with an increase in ages of raised beds the clay content of soil was increased, but sand content was decreased, implying that the soil textures have changed over the cultivation period. This finding does not align with the findings of (Quang, 2013; Murano, Takata & Isoi, 2015; Dang & Hung, 2023), who demonstrated that the age of raised beds in fruit tree orchards does not affect the soil texture, and asserted that soil texture is a permanent property unaffected by management practices. The soil texture of the raised bed was identified as silty clay and silty clay loam (Table 1). These orchard soils having an origin from alluvial island deposits with silt clay are highly suitable for fruit plants (Tran et al., 2020). An exploratory data analysis using Pearson’s Correlation revealed the relationship between the ages of raised beds and physical soil properties was significant (Table S2). Notably, a strong positive correlation was found between raised bed ages and bulk density (r = 0.666**), while negative correlations were observed with soil moisture content (r = −0.570**) and soil porosity (r = −0.666**). These findings suggested that older raised beds in Longan orchards are associated with higher degrees of soil compaction, which leads to lower soil porosity and reduced soil water content storage capacity in the rhizosphere. Several previous studies have shown that soil compaction significantly impacts soil structure, soil bulk density, penetrometer resistance, soil aeration, water infiltration, and hydraulic conductivity, and subsequently crop growth and root growth are hampered (Raper & Mac Kirby, 2006; Chan et al., 2006; Radford et al., 2007; Hula, Kroulik & Kovaricek, 2009; Horn et al., 2019; Keller et al., 2019). The adverse effects of soil compaction further result in decreased plant emergence, plant establishment, and plant height (Sidhu & Duiker, 2006; Millington et al., 2016; Shaheb, 2020). In severe cases, soil compaction significantly affects crop growth, development, yield, and farm income (Botta et al., 2010; Chamen, 2011; Godwin et al., 2017; Shaheb et al., 2018; Colombi & Keller, 2019). Therefore, implementing appropriate reclamation practices are critical if the soil’s physical environment is to be improved, thereby enhancing soil health for optimal fruit growth, yield, and quality. Dung et al. (2022); Dung et al. (2023) showed that using leguminous plants as a cover crop or rice straw mulch helped to increase the soil organic matter, soil nutrients, and porosity in citrus orchard raised beds in the Mekong Delta region of Vietnam.

Soil chemical properties of different Longan orchards

The chemical properties of different Longan orchard soils among different raised bed age groups from 15 to 60-year-old are shown in Table 2. There was a tendency of decreasing soil pH by raised bed age at 0–20 cm soil depth, however, no clear significance was found among the groups of raised bed ages (p > 0.05). The soil pH value ranged from 4.86 to 5.2, classifying it as very strongly acidic, which is considered low for agricultural soils. Soil organic matter, total phosphorus, total potassium, available P, available Si and exchangeable Ca2+ were found to differ significantly between raised bed age groups (p < 0.05). The studied soil nutrient elements strongly decreased with the age of the raised beds, with reductions in the 46–60 year raised bed group ranging from 43.76% to 63.73%. While available N (NH4+, NO3−), available B, Zn and Mn were significantly higher in young raised beds (p < 0.05). pH, EC, total N, and exchangeable K were low, Cu was below detectable range in all 20 soil samples.

Table 2 Soil chemical properties of different Longan orchards in Vinh Long, Vietnam.

Raised bed ages (Year)	Soil chemical properties	
	pHw (1:2.5)	EC (dS m −1 )	SOM (%)	Total N (%)	Total P (%)	TotalK (%)	NH4+ (mg kg−1)	NO3− (mg kg−1)	Avail. P (mg kg−1)	Exch. K+ (meq 100 g−1)	Exch. Ca2+ (meq 100 g−1)	Avail. Si (mg kg−1)	Avail. B (mg kg−1)	Avail. Zn (mg kg−1)	Avail. Cu (mg kg−1)	Avail. Mn (mg kg−1)	
15–25	5.20	0.224	4.60a	0.233	0.504a	3.43a	17.2	29.0	494.a	0.235	5.42a	11.0ab	2.41	6.83	nd	25.9	
26–37	4.76	0.324	3.97a	0.193	0.400ab	2.01b	18.4	29.5	439ab	0.328	2.45b	18.3a	2.25	5.63	nd	25.4	
38–45	4.93	0.324	3.84ab	0.217	0.403ab	1.95b	17.3	32.0	378b	0.251	3.27b	16.3a	2.05	6.31	nd	25.3	
46–60	4.86	0.218	2.61b	0.157	0.205b	1.93b	17.1	23.4	218c	0.226	2.74b	4.09b	2.13	7.89	nd	27.4	
F	ns	ns	*	ns	*	*	ns	ns	*	ns	*	*	ns	ns	ns	ns	
CV (%)	6.92	37.3	26.7	29.0	46.6	29.5	34.0	44.3	30.3	36.2	43.8	67.1	15.6	36.5	–	22.4	
Notes.

SOM is soil organic matter.

*, means in the same rows with different letters are significantly (p < 0.05) different according to Tukey’s test.

ns, non-significant, nd, not detected; n = 5.

Longan prefers soil with a pH ranging from about 6.0 to 6.5 but is tolerant of acidic soil. A range of pH values from 4.76 to 5.20 among the investigated Longan orchards may adversely affect the growth and yield of Longan. The low pH may be attributed to various factors, including plant uptake of base cations, nutrition leaching, accumulation of hydrogen ion (H+), and unbalanced fertilizer supply (Quang & Guong, 2011). The previous study revealed that farmers in this area frequently overuse inorganic fertilizers (urea, KH2PO4, (NH4)2HPO4, KCl) with large quantities of inorganic P fertilizers in their orchards to enhance fruit productivity (Le & Ngo, 2022). A low pH condition can cause a deficiency of nutrients such as calcium, magnesium, phosphorus, and molybdenum as well as disturb the mineralization and nitrification process. In the results, the older age group revealed a decrease of 50% in soil organic matter, total P, total K, available P, exchange Ca2+, and available Si compared to the younger age group, consequently leading to adverse effects on plant growth and an imbalance in soil nutrients under low pH conditions. The soil pH values of all collected soil samples in this current study were classified as ranging from low to very strongly acidic in the 0–20 cm soil depth, indicating a spectrum of acidification over the long-term period for Longan cultivation. This phenomenon is attributed to the prolonged overuse of N based chemical fertilizers by Longan farmers in the study site, aligning with the findings of Ge, Zhu & Jiang (2018).

Similar to findings by Wang et al. (2013) and Hu, Jin & Wu (2017), the elevated levels of available phosphorus, NH4+, NO3−, B, Zn and Mn in all soil samples, irrespective of the ages of raised beds, were consistent with the prolonged high application of N, P, K and other micronutrient fertilizers in the study area (Hou et al., 2021). Typically, from the household survey we can see that the rate of N chemical fertilizer applied in Longan orchards is much higher than the recommended application rate for Longan trees. This practice can lead a substantial accumulation of available nitrogen in the 0–20 cm soil depth (Wang et al., 2013). Considering the P over fertilization and its strong residual effects, the application rate of P chemical fertilizer should be reduced in most Longan orchards of the study area (Bruun, Mertz & Elberling, 2006). Local farmers have recognized that Longan trees are highly susceptible to B, Zn, and Mn deficiency based on our survey, and their increased attention to B, Zn, and Mn fertilization could explain the high B, Zn, and Mn accumulation in the topsoil of Longan orchards. Copper (Cu) is an essential element for plant growth, however, no Cu was detected in soil analyses. This suggests there may be some benefit to Longan farmers in applying Cu to their soils. The repeated high applications of urea, KH2PO4, (NH4)2HPO4, KCl as main chemical fertilizers for Longan without Ca input have caused a significant depletion of soil Ex-Ca which aligns with the study by Li et al. (2017) on the apple orchard soils. Reducing N and K chemical fertilizer inputs while manure application may serve as an alternative approach to remediate soil Ex-Ca depletion, thereby preventing bitter pit that occurs in fruit tree orchards, as suggested by Li et al. (2017).

The findings of this research are consistent with previous studies investigating the impact of raised bed age on physicochemical properties in fruit orchard soil. Research conducted on soil morphological and physico-chemical characteristics of raised-bed soil profiles under Nam Roi pomelo tree cultivation in Hau Giang province, Vietnam similarly showed that prolonged pomelo cultivation affected these soil characteristics, resulting in restricted growth and reduced yield of pomelo. Particularly, the results illustrated that low pH values (4.0–6.0), along with decreasing organic matter and available P content observed gradually changed with increasing depths (Dung et al., 2020). According to (Dang & Hung, 2023), a young bed (10 years old) and an old bed (42 years old) showed a significant difference of available P, soil organic matter, exchangeable cations, available water-holding capacity, and soil porosity with a decrease from 15% to 20% from young beds to older beds. Similarly, research on long-term cultivation of kiwi plantations found significant decreases in soil organic carbon, total N, and calcium concentrations, cation exchange capacity (CEC), water holding capacity, pH, sand content, and N mineralization and nitrification rates (Shan et al., 2020). Our findings indicate that soil organic matter, total P, total K, available P, exchangeable Ca2+, and available Si were significantly reduced with increasing ages of Longan cultivation. Similarly, many other previous studies on a county-scale spatial variability of soil pH and SOC, as well as that of available N, P, and K (0–40 cm depth), have consistently reported low levels of soil organic matter and other chemical properties under long-term apple cultivation (Guo et al., 2015; Zhang et al., 2016). Moreover, the relatively low levels of soil organic matter detected in this study can be attributed to several factors, including the highly weathered nature of soils, the warm and humid climate, and the consequent accelerated breakdown of soil organic matter by soil microbes (Slade & Wells, 2022).

Comparisons between soil chemical properties and the ages of raised beds of Longan using Pearson correlation analysis showed s showed that only soil organic matter and soil total N revealed a correlation with raised bed age (Table S3). Specifically, the soil organic matter indicated a strongly negative relationship with raised bed age (r = −0.579**, p < 0.01), with the age of raised beds in Longan orchards accounting for 34% of the variation in soil organic matter measurements (R2 = 0.34). Similarly, soil total N also exhibited a highly negative relationship with raised bed age (r = −0.47* (p < 0.05)). The raised bed ages of Longan orchards accounted for 22% of the variation in total N measurements (R2 = 0.22). This finding indicated that the long-term use of raised beds for cultivating Longan caused a significant reduction in the content of soil organic matter and total N. Therefore, there is a need for a more sustainable management approach for aged Longan orchards in Vinh Long, VMD, to overcome the low soil fertility and quality, to secure the Longan yield and incomes of Longan farmers.

Biological parameters of different Longan orchard soils

Microbiological densities of different Longan orchard soils

Counts of different groups of soil bacteria in different Longan orchards are illustrated in Table 3. The results showed that among the five surveyed bacterial groups, nitrogen-fixing bacteria had the highest density across the four different raised bed age groups. This was followed by phosphorus solubilizing-bacteria and potassium solubilizing bacteria, while calcium solubilizing bacteria, and silicate solubilizing bacteria were not detected in any of the studied orchard soils. Among them, only nitrogen-fixing bacteria had significant differences between the raised bed age groups. In particular, the 15–20 years old group was significantly highest at 6.26 (log10 cfu g−1) (p < 0.05) while the other three groups (26–37; 38–45, and 46–60 years old) were significantly lower (p < 0.05), but they did not differ from each other (p > 0.05), ranging from 5.73 to 5.88 (log10 cfu g−1). This may relate to the amount of organic matter in the soil. A study by Hai et al. (2009) showed that after applying organic fertilizer (organic fertilizer and straw), the nitrogen-fixing population was the highest dominant position in the investigated community in soil and the soil organic amount of the soil group containing 15–25 years of raised bed was the highest at 4.6%. Thus, organic matter content in soil is one of the soil indicators for high population of nitrogen fixing bacteria in soil. Counts of phosphorus solubilizing bacteria and potassium solubilizing bacteria were not statistically different between bed age groups, ranging from 4.57 to 5.06 (log10 cfu g−1) and from 2.57 to 3.09 (log10 cfu g−1), respectively. The counts of bacteria regardless of bacterial types were reduced with increasing raised bed age. This might be because of the severely acidic soil conditions (low soil pH) (Table 3) in aged Longan orchards, which inhibited the growth and enrichment of soil bacteria, especially nitrogen fixing bacteria, resulting in a reduction in the number of bacteria. This study is consistent with the previous studies by Rousk et al. (2010); Lin & Wang (2012) and Fu et al. (2015) who also studied about the soil microbial numbers in aged kiwifruit orchards and they found that the numbers of soil cultivable microorganism decreased with the increasing of aged kiwifruit orchards. Longan tree is a kind of perennial crop and Longan orchard soils are usually managed as a monoculture. Over time as Longan trees become older, farming and soil management practices become relatively fixed. The traditional method of soil management which farmers in this study site have adopted has led to a gradual deterioration of the soil micro-ecological environment, resulting in the reduction of soil microbial activity, and the loss of soil microbial functional and structural diversity. Ultimately, this would restrict the uptake and use efficiency of nutrients from soil and consequently affect the yield and quality of the Longan. However, there is a need for additional study to better understand the impact of aged Longan orchards on soil microbial community structure and its ecological function using other molecular biological tools and techniques.

Table 3 Microbiological densities of different Longan orchard soils in the Mekong River Delta of Vietnam.

Raised bed age (Year)	Soil microbial numbers (Log10 cfu g−1)	
	Nitrogen fixing bacteria	Phosphorus solubilizing bacteria	Potassium solubilizing bacteria	Calcium solubilizing bacteria	Silicate solubilizing bacteria	
15–25	6.26a	5.06	3.09	nd	nd	
26–37	5.73b	4.89	2.57	nd	nd	
38–45	5.88b	4.73	2.96	nd	nd	
46–60	5.83b	4.57	2.62	nd	nd	
F	*	ns	ns	–	–	
CV (%)	4.14	9.04	14.3	–	–	
Notes.

* Means in the same rows with different letters are significantly (p < 0.05) different according to Tukey’s test; ns, non-significant, nd: not detected; n = 5.

Soil enzyme activities of different Longan orchard soils

The results presented in Table 4 show the activities of selected soil enzymes in Longan orchards of four different raised bed ages. Data reveal a significant pattern of decreased enzyme activity with increasing age of the raised beds among the 4 Longan orchard soil groups. Notably, the β-glucosidase enzyme activity decreased tenfold as the raised bed age increased from 15–20 years to 50–60 years, while the activities of three other enzymes decreased by half. Obviously, the younger the raised bed age, the higher enzyme activity in the soil. Our results showed that all four studied enzyme activities including β-glucosidase, urease, phosphomonoesterase, and phytase significantly decreased with the increasing age of the Longan orchards (Table S4). This could be due to the soil organic matter decreasing also with the age of Longan orchards (Table 2). This result aligns with the findings of Rakshit et al. (2018), which demonstrate that organic matter acts as a substrate for soil enzymes. Consequently, soil enzymes are synthesized by soil microbes and their activity is strongly regulated by the presence of organic matter, serving as a source of soil organic carbon. In a study by (Dang & Hung, 2023), it was found that soil pH, electrical conductivity, available phosphorus, total nitrogen, soil organic matter, exchangeable cations (K+, Na+,…), cation exchange capacity, bulk density, and soil porosity of the older Longan orchards significantly decreased. Additionally, perennial Longan gardens exhibited low ventilation and high compaction which also resulted in reduced enzyme activity.

Table 4 Soil enzyme activities of different Longan orchard soils in the Mekong River Delta of Vietnam.

Raised bed age (Year)	Soil enzyme activities (µg g−1 h−1)	
	B-glucosidase	Urease	Phosphomonoesterase	Phytase	
15–25	33.4a	32.9a	32.9a	7.41a	
26–37	13.3c	18.1bc	18.1bc	5.66b	
38–45	29.4b	23.8b	23.8b	5.59b	
46–60	3.31d	16.5c	16.5c	4.92b	
F	*	*	*	*	
CV (%)	63.5	32.6	43.4	21.3	
Notes.

* Means in the same rows with different letters are significantly (p < 0.05) different according to Tukey’s test; n = 5.

The β-glucosidase enzyme, which is widely distributed in the soil, plays a crucial role in the carbon cycle, by cleaving of cellobiose into glucose molecules. This activity is closely associated with the quantity and quality of soil organic matter (Almeida, Naves & Da Mota, 2015). Urease is the hydrolytic enzyme that degrades urea, and is widely regarded as an effective proxy for nitrogen (N) mineralization in soil (Cordero, Snell & Bardgett, 2019). Phosphomonoesterase, and phytase are specific enzymes that mineralize organic phosphorus from phosphomonoesters, and inositol phosphates, respectively (Turner, 2008). Our study indicates that the changes in the activity levels of soil enzymes- β-glucosidase, urease, and phosphomonoesterase, and phytase-followed the same trends as soil organic matter, total N, and total P, respectively. This is likely because these enzymes are readily induced by their respective substrates. These results suggest that the activities of these enzymes are highly dependent on soil organic matter, total nitrogen and total phosphorus in soil, consistent with the findings of (Turner, 2008; Nannipieri et al., 2012). The activities of these soil enzymes, along with soil organic matter, total nitrogen and total phosphorus exhibited a significant decrease with the increasing age of the Longan orchards. The highest enzyme activities were observed in the younger raised beds of Longan orchards, indicating robust soil nutrient mineralization and ample availability of soil nutrients in these younger beds. During the initial phase of raised bed cultivation, Longan trees exhibit vigorous growth. Further nutrient accumulation in the fruit demands an adequate supply of available nutrients. This phenomenon may be attributed to the trees releasing root exudates, which stimulate the soil microbial activity in the rhizospheric zone. Consequently, the trees can enhance the uptake of limiting soil nutrients that may otherwise be limited (Hamilton III & Frank, 2001). A previous study by Xue, Yao & Huang (2006) on tea orchard soil in China reported that urease enzyme activity was notably higher in younger tea orchards, whereas in the aged tea orchards (>50 years old) it was significantly lower. Further investigations by Dong et al. (2009) and Zhou, Zhang & Downing (2012) revealed that even in younger tea orchards (6 years old) urease enzyme activity strongly decreased (Dong et al., 2009; Zhou, Zhang & Downing, 2012).

In our current study, we observed a significant positive correlation between the β-glucosidase enzyme and respective soil parameters such as soil organic matter ( r = 0.92 ***, R2 = 0.846), total nitrogen (r = 0.76***, R2 = 0.578), exchangeable Ca2+ (r = 0.57 **, R2 = 0.33), soil moisture content (0.74***, R2 = 0.55), soil particle density (r = −0.73***, R2 = 0.53), and soil porosity (r = 0.52*, R2 = 0.27) (Fig. 2). Previous research has indicated that the primary direct driver of β-glucosidase enzyme in soil is organic matter content (Stott et al., 2010; Miralles et al., 2012). Furthermore, β-glucosidase activity in the soil has been found to be sensitive to a range of soil management practices and different soil types and textures. In degraded soils with low organic matter content, there is a reduction in β-glucosidase activity, leading to a diminished availability of simple sugars for microbial populations (Stott et al., 2010). Soil moisture content also regulates the activity of β-glucosidase enzymes in soil and influences the biochemical processes of soil C transformation (Zhang et al., 2011). Mulching with organic materials, such as crop residues, on the soil surface is a proper management practice for conserving soil moisture content and temperature, and can lead to an increase in β-glucosidase enzyme activity (Sarkar, Paramanick & Goswami, 2007; Murungu et al., 2011; Zhang et al., 2011; Mukumbareza, Muchaonyerwa & Chiduza, 2015; Adetunji et al., 2020). Zhang et al. (2007) found that mulching in dryland soils enhanced the water holding capacity by up to 8%, reducing soil evaporation by around 13%. Moreover, Almeida, Naves & Da Mota (2015) showed that the activity of the β-glucosidase enzyme is not only associated with the accumulation of soil organic content but also with the quality of vegetation residues and litter. The relationship between β-glucosidase enzyme and other soil particle density and soil porosity may be indirectly linked to soil organic matter. Moreover, β-glucosidase enzyme activity tends to increase with exchangeable Ca concentration, suggesting that soils developed from limestone parent material favor β-glucosidase enzyme activity. This can be attributed to calcium’s role in enhancing soil structure through soil aggregation and stimulating microbial activity (Arenberg & Arai, 2019).

Figure 2 PCA of soil physio-biochemical parameters of different Longan orchard soils in the Mekong River Delta of Vietnam.

The urease enzyme in soil is closely linked to nitrogen content and mineralization, as evidenced by numerous previous studies. For instance, Solangi et al. (2024) conducted a study on agricultural soil in China, revealing significant correlations between urease activity and microbial biomass phosphorus (r = 0.743**), available phosphorus (r = 0.498*), and a negative relationship with available K (r = − 0.686**). Additionally, urease activities in soils are associated with various soil properties such as organic matter content, texture, pH, silt, clay, sand contents, total nitrogen, and cation exchange capacity (CEC) (Speir, Pansier & Cairns, 1980; Dash et al., 1981; Reynolds, Wolf & Armbruster, 1985; Singh, Kumar & Dahiya, 1991; Dharmakeerthi & Thenabadu, 1996; Özdemir, Kızılkaya & Sürücü, 2000; Zhang et al., 2014). However, in acidic tea soils, no relationship between urease activity and OC or texture was found to be dependent on polyphenol content of soil (Wickremasinghe, Sivasubramaniam & Nalliah, 1981).

Our results showed no correlations between urease and soil organic matter, nitrogen, phosphorus and other properties, but a positive correlation with exchangeable Ca (r = 0.47*, R2 = 0.22) (Fig. 2) was observed. This suggests that multi-enzyme activity may be better correlated with soil fertility than a single enzyme (Dick & Tabatabai, 1992). These findings are consistent with those of Xue, Yao & Huang (2006) which showed no correlations between urease, proteinase and organic matter in tea orchards of different ages in China. It is well known that urease is primarily involved in the transformation of soil N. Our study demonstrates that long-term heavy chemical fertilizer application in Longan orchard soil ecosystems do not affect the urease enzyme activity involved in soil N cycles.

Phosphomonoesterase and phytase are two enzymes involved in the mineralization of organic phosphorus into inorganic phosphorus in soil. While there is no information regarding the variation of these enzyme activities with different orchard soil ages, previous studies have shown that phosphomonoesterase enzyme activity decreases with depth down to 1 m. This decline is associated with reduced fine-root mass density (Cabugao, 2020), a lack of substrate availability (Stone & Plante, 2014) and shifts in microbial biomass C (Hou et al., 2015). In our current study, phosphomonoesterase activity was positively correlated with soil moisture (r = 0.75***, R2 = 0.56), soil porosity (r = 0.63**, R2 = 0.46), soil organic matter (r = 0.85***, R2 = 0.72), total nitrogen (r = 0.68***, R2 = 0.46), total phosphorus (r = 0.47*, R2 = 0.22), and exchangeable Ca (r = 0.45*, R2 = 0.20). Conversely, it was negatively correlated with soil bulk density (r = −0.68**, R2 = 0.46), and soil particle density (r = −0.72***, R2 = 0.52). These physical and chemical soil properties can be used to predict phosphomonoesterase activity and aligns with Cabugao et al. (2021), findings of positive correlations between soil moisture content, total soil P, and phosphomonoesterase activity. Additionally, in our study, phytase activity was significantly correlated with total phosphorus (r = 0.70***, R2 = 0.49), available phosphorus (r = 0.64**, R2 = 0.41), soil moisture ( r = 0.46*, R2 = 0.21), soil porosity (r = 0.53*, R2 = 0.28), and negatively with soil bulk density (r = −0.53*, R2 = 0.28). Previous studies have shown that phytase activity was positively correlated with soil moisture, with optimal activity observed at 100% saturation, highlighting the importance of soil moisture for phytase production and microbe survival (Criquet et al., 2004; Arenberg & Arai, 2019). Furthermore, George et al. (2007) found that soil pH and temperature are critical factors regulating phytase activity. Our findings suggest that phytase activity by soil microbes can be optimized by increasing organic carbon substrate, pH, temperature, and total nitrogen (Sadaf et al., 2022).

Collectively integrated interaction among Longan soil properties, raised bed age, Longan tree age, and Longan yield

PCA analysis results showed that most indicators, including SOM, PSB, and PoSB, are correlated with Dim 2. The antagonistic relationship between bulk density, soil moisture, and porosity in Dim 2 suggested that higher bulk density leads to more aggregate compaction, reducing soil porousness (Fig. 2). Additionally, higher NO3− content in Dim 1, which is plant-accessible, was associated with higher yields in Longan trees, consistent with Cela et al. (2013) who demonstrated that soil-available-N content could accurately predict relative corn yield under irrigated, high-yielding conditions. Therefore, these findings suggest that the recommended chemical fertilizer formula for Longan needs updated, as soil properties and soil fertility have significantly changed over the past 20 years, while farmers still adhere to the outdated recommendations. This update will assist in reducing the over-application of nitrogen fertilizers, which can adversely impact the environment and human health. Notably, higher uptake of exchangeable K+ has an antagonistic relationship with exchangeable Ca2+, contributing to Dim 1 (Fig. 2). In Dim 1, NFB, exchangeable Ca2+, pHw, andavailable B are negatively correlated with yield, NO3−, EC, exchangeable K+, and available Mn and Zn. This aligns with studies showing that available nitrogen, particularly NO3−, is antagonistic with NFB, where mineral N fertilizer application suppresses N-fixing bacteria and enriches disease-related microbial functions (Lekberg et al., 2021; Zhou et al., 2021; Wassermann et al., 2023). The higher soil pH associated with exchangeable Ca2+ may not be ideal for Longan, which prefers slightly acidic soil. High available B content can cause phytotoxicity, despite boron (B) being an essential micronutrient (Miwa & Fujiwara, 2010; Shireen et al., 2018; Wang et al., 2024) but becomes toxic in higher quantities in soil for Longan trees. Therefore, Longan farmers should avoid continuous addition of B into their soils. Integrating soil biological parameters and enzyme activities reveals crucial insights into Longan orchard soils. Microbial counts, particularly NFB and PSB, exhibit notable correlations with SOM and phosphorus content, highlighting the role of microbial communities in nutrient cycling and availability. Enzyme activities, such as β-glucosidase, urease, phosphomonoesterase, and phytase, decrease with increasing raised bed age, indicating gradual degradation of soil organic matter and nutrient turnover due to compaction and reduced porosity. This underscores the importance of soil management practices to sustain optimal soil functioning for Longan cultivation. The PCA results illustrate the interconnectedness of biological parameters and enzyme activities with soil physicochemical properties, providing a holistic perspective on soil health and its implications for Longan orchard management.

Pearson’s correlation analysis (Fig. 3) confirms the relationships between the main soil properties indicated by the PCA. Correlation coefficients between variables and their significance were used to evaluate soil health and fertility following Longan tree cultivation and to specify their significant relationships. Notably, significant positive correlations were observed between soil moisture and total P (p < 0.01) and N (p < 0.001), available P (p < 0.05), and NH4 (p < 0.05). These findings reflect the role of well-moisturized soils in enhancing the solubility and mobility of soil nutrients, thereby elevating nutrient absorption by plant roots (Wang et al., 2023). Additionally, SOM was significantly correlated with increases in soil nutrients, including P (p < 0.05) and total N (p < 0.001), and NH4+ (p <  0.05). Other soil physical properties, such as soil porosity, pH, and EC, significantly influenced the availability of N, P, and B, as well as the exchange of Ca. In contrast, direct negative correlations were observed between the soil bulk density and soil porosity (p < 0.001), organic matter (p <  0.001), N (p < 0.001), and total P (p <  0.01). The availability of P (p < 0.05), and the abundances of soil microbes, including PSB populations (p < 0.05), were consistent with previous studies (Zhang et al., 2022; Zhou et al., 2024).

Figure 3 Correlation matrix reveals the relationships among soil physical, chemical, and biological properties as well as the agronomic traits of Longan tree.

The status of organic matter in the soils due to Longan tree cultivation is a crucial factor in enhancing the abundance of key soil microorganisms, such as PSB (p < 0.05), which play vital roles in nutrient cycling. Additionally, the activity of phosphomonoesterase exhibited a strong correlation with SOM (p < 0.001), ultimately increasing soil available P. This scenario suggests that the Longan tree cultivation provides essential nutrients and energy sources, and improves soil structure and creates a favorable environment for microbial activities, including PSB (Bünemann, Prusisz & Ehlers, 2011). Increased SOM was also significantly correlated with the content of ß-glucosidase, an enzyme critical for degradation of complex carbohydrates into glucose (Yang et al., 2023). However, negative correlations were found between the age of the raised beds in logan tree cultivation and soil porosity (p < 0.01), SOM (p < 0.05), and N total (p < 0.05). This may be attributed to management practices such as regular tillage over time, which can increase soil compaction and deplete organic matter, leading to decreased soil porosity and total N (Iqbal et al., 2021; Dung et al., 2022). Despite these challenges, Longan yields significantly benefit from other soil properties, such as soil available Zn. This essential nutrient regulates growth hormones and enzymes, influencing metabolic processes including photosynthesis and protein synthesis (Broadley et al., 2007; Saleem et al., 2022).

Conclusions

Over time, the soil in the raised beds of Longan orchards in the Mekong Delta can lose key chemical and physical properties that are sources of important nutrients needed to achieve high yields. The novelty of this research is the roles soil microbiological properties and enzyme activities playing in nutrient and soil fertility in Longan soils and the effects of time on the soil of trees grown in raised beds over 50–60 years. We find that soil microbial properties and soil enzyme activities have synergistic potential to stimulate populations of beneficial soil bacterial communities that can restore the soil health of aging Longan orchards. The soil quality indicators measured in this study show Longan orchards in raised beds in Vinh Long province, Vietnam under conventional management practices have significantly increased soil bed degradation and soil compaction over the last 60 years. Data trends show lower levels of soil moisture, soil porosity, and sand content, in aging beds. As the Longan orchard’s raised beds aged, soil fertility decreased including soil organic matter, total phosphorus, total potassium, available phosphorus, exchangeable Ca2+, and available Si; and reduced biological activity occurred such as nitrogen fixing bacteria, ß-glucosidase, urease, phosphomonoesterase, and phytase. Raised bed ages of Longan orchards were strongly correlated with soil bulk density, soil moisture, soil porosity, soil organic matter, total nitrogen, ß-glucosidase, urease, phosphomonoesterase, and phytase, which explained a great deal of soil property variations observed in the raised bed ages. Longan raised beds are positively impacted by higher soil available NO3− and Zn, but negatively impacted by higher levels of exchangeable Ca 2+, pH and soil available B. Soil moisture content, soil porosity, soil organic matter, total nitrogen, and total phosphorus have positive correlations with beneficial soil microorganism such as phosphorus solubilizing bacteria, and soil enzyme activities including ß-glucosidase, phosphomonoesterase, and phytase while soil bulk density and soil particle density have negative correlation with these soil microbial properties.

This research shows that soil biological, chemical and physical properties and soil fertility of these orchard soils have changed greatly over the last 20 years. Farmers who are following out-of-date recommended chemical fertilizer formulas are at risk of over applying chemical fertilizers, especially nitrogen which can leach off the raised beds which is a costly loss of nutrients, and adverse impacts on the environment and human health. There is a clear need for additional research on optimizing and updating the chemical fertilizer formula recommendations for the Vietnam Mekong Delta orchard production based on current soil conditions and identifying management practices that can improve Longan production in this region.

Future research should build on these findings and expand soil microbial work on the effects of soil bio-stimulation approaches such as applications of soil amendments, organic materials, micronutrients, cover crops and earthworms. Additional work on soil beneficial bacteria and improving productivity of agricultural soils should include nitrogen fixing bacteria, phosphorous solubilizing bacteria, potassium solubilizing bacteria, nutrient related enzyme activities and other soil physical and chemical properties that affect the field conditions and production of VMD orchards.

Supplemental Information

Supplemental Information 1 Raw data

Supplemental Information 2 Supplemental Tables

Abbreviations

ANOVA analysis of variance

AP available phosphorus

B boron

Ca2+ exchangeable calcium

CEC cation exchange capacity

Cu copper

DB devenze-Bruni

EC electrical conductivity

H+ hydrogen ion

K+ exchangeable potassium

Mn manganese

N nitrogen

NBRIP national Botanical Research Institute’s phosphate growth medium

NFB nitrogen-fixing bacteria

PCA principal component analysis

PME phosphomonoesterase

PoSB potassium solubilizing bacteria

PSB phosphate saline buffer

PSB phosphorus solubilizing bacteria

SD standard deviation

SEA soil extract agar

SiO2 available silicate

SOM soil organic matter

TK total potassium

TN total nitrogen

TP total phosphorus

VDM Vietnamese Mekong Delta

Zn zinc

Additional Information and Declarations

Competing Interests

Author Contributions

Data Availability

The authors declare there are no competing interests.

Nghia Khoi Nguyen conceived and designed the experiments, authored or reviewed drafts of the article, and approved the final draft.

Phuong Minh Nguyen conceived and designed the experiments, analyzed the data, authored or reviewed drafts of the article, and approved the final draft.

Anh Thy Thi Chau conceived and designed the experiments, analyzed the data, authored or reviewed drafts of the article, and approved the final draft.

Luan Thanh Do conceived and designed the experiments, performed the experiments, analyzed the data, prepared figures and/or tables, and approved the final draft.

Thu Ha Thi Nguyen conceived and designed the experiments, performed the experiments, analyzed the data, prepared figures and/or tables, chemical, reagent, and approved the final draft.

Duong Hai Vo Tran conceived and designed the experiments, performed the experiments, prepared figures and/or tables, authored or reviewed drafts of the article, and approved the final draft.

Xa Thi Le conceived and designed the experiments, performed the experiments, prepared figures and/or tables, authored or reviewed drafts of the article, and approved the final draft.

Javad Robatjazi analyzed the data, prepared figures and/or tables, authored or reviewed drafts of the article, and approved the final draft.

Hendra Gonsalve W. Lasar analyzed the data, prepared figures and/or tables, and approved the final draft.

Lois Wright Morton conceived and designed the experiments, analyzed the data, authored or reviewed drafts of the article, and approved the final draft.

M. Scott Demyan conceived and designed the experiments, analyzed the data, authored or reviewed drafts of the article, and approved the final draft.

Huu-Tuan Tran conceived and designed the experiments, analyzed the data, prepared figures and/or tables, authored or reviewed drafts of the article, and approved the final draft.

Hüseyin Barış Tecimen conceived and designed the experiments, analyzed the data, prepared figures and/or tables, authored or reviewed drafts of the article, and approved the final draft.

The following information was supplied regarding data availability:

The raw data is available in the Supplementary File.

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
