# Peer review of "Long-term changes in soil biological activity and other properties of raised beds in Longan orchards"

_PeerJ, doi:10.7717/peerj.18396_

## Round 0.1 · original submission · Major Revisions

Dear Dr. Nguyen

Thank you for your submission to PeerJ.

It is my opinion as the Academic Editor for your article - Changes in raised bed soil biological activity and other soil properties of Longan orchards over long time periods in the Vietnamese Mekong Delta - that it requires a range of major and minor changes.

This decision is based on the perusal of review reports. You are accordingly advised to critically revise your manuscript, keeping each and every suggestion by the reviewers in mind. It is important to mention that your revised manuscript will undergo additional peer review in order to ensure that it is suitable for publication.

Hope to receive the revised manuscript in due course.

**Language Note:** The review process has identified that the English language must be improved. PeerJ can provide language editing services - please contact us at [email protected] for pricing (be sure to provide your manuscript number and title). Alternatively, you should make your own arrangements to improve the language quality and provide details in your response letter. – PeerJ Staff

Reviewer 1 ·

Basic reporting

Dear Author,
Thank you for providing the opportunity to review the manuscript entitled “Changes in raised bed soil biological activity and other soil properties of Longan orchards over long time periods in the Vietnamese Mekong Delta”. Some major mistakes were found which need to be rectified, following are the comments mentioned below.
1. Write the keywords in alphabetical order.
2. In line no. 65 write "high" market demand.
3. Line No. 66-70, reframe the sentence.
4. Line No. 74 the references mentioned in the MS is not according to the journals format.
5. In line no. 75 remove comma present in between the numbers. write as 1370-2394 mm
6. Line No. 82-85, reframe the lines.
7. As authors mentioned that similar research has been performed by Xiang et al., 2018; why authors perform the same kind of research again? Authors are suggested to highlight the novelty of the present research clearly.

Experimental design

1. Authors are suggested to improve English language to ensure that others can clearly understand the text.
2. In line no 240 write 'data analysis' instead of analyzing the data.
3. Line no. 330-341, make it concise and to the point.
4. Most of the lines and statements used in the manuscript are 'general'. Authors are suggested to remove all the general statements from the manuscript. For example, Line No. 397-400; 432-433; these kinds of general statements should not be written in scientific research paper.
5. Authors are suggested to remove all the references and related statements/lines from the conclusion part and can be added in the discussion part if necessary without repeating it.

Validity of the findings

The novelty of the study is not stated clearly.
The references mentioned are not according to the journals format. Kindly check the journals format and write correctly.

Additional comments

NA

·

Basic reporting

In my opinion, this study is an important contribution to the field of fruit sciences. However, I have observed many drawbacks that should be rectified before the paper is accepted for publication

Experimental design

methodology and experimental design is adequate, however for PCA analysis pl. refer given comments in attachment

Validity of the findings

statistical analysis of data is appropriate, and the conclusion is appropriate and supported by logical reasoning

Additional comments

pl. see attached file

---

## Round 0.2 · accepted · Accept

Dear Dr. Nguyen

Thank you for your submission to PeerJ.

I am writing to inform you that your manuscript - Long-term changes in soil biological activity and other properties of raised beds in Longan orchards - has been Accepted for publication. Congratulations!

This is an editorial acceptance; publication is dependent on authors meeting all journal policies and guidelines.

Reviewer 1 ·

Basic reporting

Dear Author,
Thank you for providing the opportunity to review the manuscript entitled “Long-term changes in soil biological activity and other properties of raised bed in Longan orchards”. The references are not according to the journals format throughout the MS. Kindly check the journals format and write correctly. For example, (Raper & Kirby, 2006) and (Eivazi and 252 Tabatabai, 1988) both the references are in different format. Also the journal names are written in full or in abbreviation, maintain the uniformity. In some places the volume and issue number are Italics. Kindly go through it for a proper uniformity.

Experimental design

NA

Validity of the findings

NA

Additional comments

NA